# Clinical Efficacy and Openness to New Challenges of Low Dose Rate Brachytherapy for Prostate Cancer

Manabu Kato [1,*] , Shinichiro Higashi [2], Yusuke Sugino [2] , Shinya Kajiwara [2], Shiori Tanaka [1], Goshi Kitano [1], Yasuhumi Yamashita [3], Yuji Ogura [4], Hiroyuki Tachibana [1], Takahiro Kojima [1] and Takahiro Inoue [2]

[1] Aichi Cancer Center, Urology, Nagoya 464-8681, Japan; s.tanaka@aichi-cc.jp (S.T.); g.kitano@aichi-cc.jp (G.K.); tchbn@aichi-cc.jp (H.T.); t.kojima@aichi-cc.jp (T.K.)
[2] Department of Nephro-Urologic Surgery and Andrology, Mie University Graduate School of Medicine, Tsu 514-0001, Japan; s-higashi@med.mie-u.ac.jp (S.H.); y-sugino@med.mie-u.ac.jp (Y.S.); s-kajiwara@med.mie-u.ac.jp (S.K.); tinoue28@med.mie-u.ac.jp (T.I.)
[3] Matsusaka Chuo General Hospital, Radiology, Matsusaka 515-0818, Japan; yamasita@clin.medic.mie-u.ac.jp
[4] Kuwana City Medical Center, Urology, Kuwana 511-0061, Japan; yogura@kuwana.or.jp
[*] Correspondence: man.kato@aichi-cc.jp

**Abstract:** Over a century ago, low-dose-rate (LDR) brachytherapy was introduced to treat prostate cancer (PCa). Since then, it has been widely applied worldwide, including in East Asia. LDR brachytherapy has been performed in 88 institutes in Japan. Beneficial clinical outcomes of LDR brachytherapy for intermediate-to-high-risk PCa have been demonstrated in large clinical trials. These clinical outcomes were achieved through advances in methods, such as urological precise needle puncture and seed placement, and the quantitative decision making regarding radiological parameters by radiation oncologists. The combined use of LDR brachytherapy with other therapeutic modalities, such as external beam radiation and androgen deprivation therapy, for the clinical risk classification of PCa has led to better anticancer treatment efficacy. In this study, we summarized basic LDR brachytherapy findings that should remain unchanged and be passed down in urology departments. We also discussed the applications of LDR brachytherapy for PCa in various clinical settings, including focal and salvage therapies. In addition, we highlighted technologies associated with brachytherapy that are under development.

**Keywords:** prostate cancer; brachytherapy; low-dose-rate brachytherapy; tri-modality therapy; salvage brachytherapy; focal brachytherapy



## 1. Introduction

The prevalence of prostate cancer (PCa) in men is increasing worldwide owing to the aging population and the widespread screening for prostate-specific antigen (PSA); these factors have resulted in PCa having the highest recorded morbidity rate among all male-related malignancies [1]. In 2015, PCa was the leading type of male-related cancer, followed by stomach and lung cancers [2]. This trend has also been observed in Japan.

Surgical intervention, external beam radiotherapy, proton therapy, heavy ion radiotherapy, low-dose-rate (LDR) brachytherapy, high-dose-rate brachytherapy, and active surveillance are generally used to treat localized PCa (https://www.auanet.org/guidelines-and-quality/guidelines/clinically-localized-prostate-cancer-aua/astro-guideline-2022 (accessed on 4 November 2023)) [3]. Among these treatment options, LDR brachytherapy is preferred for patients with PCa who are older or have poor tolerance to treatment. Barringer invented and reported this method for the first time in 1917 (Barringer BS. Radiation for the treatment of bladder and prostate carcinomas JAMA 1917; 68:1227–1230). Since then, the procedure for LDR brachytherapy has been improved, and sophisticated methodologies have been introduced over the years, leading to the procedure being widely used worldwide with good clinical outcomes [4,5]. Barringer predicted that surgery for PCa

could become extinct in the future. This claim has not been in the spotlight for many decades; however, with the advent of tri-modality therapy comprising hormone therapy, brachytherapy, and external beam radiotherapy for high-risk PCa and its favorable clinical outcomes, it may be a viable goal.

In the first part of this study, we show the clinical outcomes of LDR brachytherapy in multiple institutes in Japan. In the second part, we describe the clinical outcomes in patients with localized low-to-intermediate PCa. In the third part, we discuss clinical trials investigating multi-modal brachytherapy for localized intermediate-to-high-risk PCa in Japan. Next, we summarize implantation methods and seed types of brachytherapy in detail. Finally, we described new, challenging procedures in LDR brachytherapy, including tri-modality therapy, salvage brachytherapy, focal therapy, and future automatic brachytherapy equipment development.

Regarding the invariable indications for LDR brachytherapy, most urologists consider that patients with low or intermediate PCa should undergo LDR brachytherapy alone or combined with external beam radiation therapy, which can be observed in recent clinical case conferences held in nearly every urology department. Even with the recent technological advancements, not many high-risk PCa cases are treated with tri-modality therapy. Furthermore, radiation oncologists and urologists who are not performing LDR brachytherapy tend to dwell on the misalignment of LDR seeds unnecessarily. This leads to suspicion among the other radiation oncologists and urologists regarding the precision of the brachytherapy performed. In addition, the reputation of brachytherapy as a monotonous procedure performed by a single urologist in every institute may hinder the widespread applicability of the procedure. We hope that the present study will aid in resolving these concerns, as it summarizes not only the invariable factors that lead to good clinical outcomes but also the importance of openness to new challenging procedures of LDR brachytherapy for PCa.

## 2. Materials and Methods

To perform this study on current LDR brachytherapy for prostate cancer, our group conducted a comprehensive search using different databases. These different databases included PubMed, Springer, Elsevier Science Direct, and Web of Science. The publication years of these original articles from different databases range from 2000 to 2023, taking into consideration their validity. The selected papers were only those reported in English, and the searches were conducted using the following keywords: "prostate cancer", "low-dose-rate brachytherapy", "brachytherapy for prostate cancer", "tri-modality therapy for prostate cancer", "LDR brachytherapy for intermediate risk prostate cancer", "LDR brachytherapy for high-risk prostate cancer", "salvage brachytherapy for prostate cancer", "focal brachytherapy for prostate cancer", and "image-guided instruments for prostate brachytherapy". Duplicate papers were excluded. During the process of checking the contents of the papers, they were screened to exclude papers unrelated to this review, and reports that were not full text were also excluded. There were several criteria for inclusion. The emphasis was on original articles of clinical research about LDR brachytherapy for prostate cancer and review articles. The exclusion criteria mainly focused on different aspects of LDR brachytherapy for prostate cancer, articles containing information unrelated to LDR for prostate cancer, and reports that could not be reached as full-text articles.

## 3. Clinical Outcomes of Localized Low-to-Intermediate PCa Treatment

Brachytherapy is considered an effective treatment option for patients with localized PCa. (https://www.auanet.org/guidelines-and-quality/guidelines/clinically-localized-prostate-cancer-aua/astro-guideline-2022 (accessed on 4 November 2023)) The American Brachytherapy Society has stated that brachytherapy is a convenient, effective, and well-acceptable treatment for localized PCa [6]. However, despite its advantages, brachytherapy cannot be performed in all clinical institutes, leading to the impression of this treatment as a "minor" option. One large review from Japan showed a median follow-up duration of

75 months and 7-year biochemical recurrence-free survival rates of 98%, 93%, and 81% in patients with low-, intermediate-, and high-risk PCa, respectively [5,7]. The indications for combination therapy in the study were as follows: low-, intermediate-, and high-risk PCa should be treated with brachytherapy alone, brachytherapy combined with irradiation therapy, and brachytherapy combined with neoadjuvant androgen deprivation and external radiation therapy, respectively [5,7]. The clinical outcomes of localized low-to-intermediate PCa treatment in our institutes are consistent with the general outcomes of brachytherapy for low-to-intermediate PCa, as reported in these reviews [5,7]. Therefore, the high biochemical recurrence-free survival rate indicates that low-to-intermediate-risk PCa could be controlled using LDR alone or LDR combined with external radiation therapy under appropriate treatment selection. In addition, most Japanese institutes employ radiation oncologists to decide the seed implant position before initiating needle puncture. The seeds are repositioned during the puncture and implantation procedures. This dynamic dose calculation method might produce sufficient dose distribution and good clinical outcomes [8] compared to other procedures in which urologists puncture the prostate before planning seed placement, a process mainly led by radiation oncologists [9]. One randomized study (RTOG0232) for intermediate-risk PCa, particularly in the GS6 and PSA 10–20, and the GS7 and PSA < 10 groups, found no significant difference in biochemical recurrence-free survival (biochemical failure) between brachytherapy combined with external beam radiation therapy and brachytherapy alone (88.0% vs. 85.5% at 5 years). However, there was a significant difference in grade 2 or higher genitourinary (GU) or gastrointestinal (GI) toxicities (42.8% vs. 25.8% in G2, 8.2% vs. 3.8% in G3 at 5 years) [10]. As mentioned above, despite some unpopularity and limited institutional procedures, strong evidence of the unchangeability and stability of brachytherapy for low-to-intermediate-risk PCa has been demonstrated.

## 4. Clinical Trial of Multi-Modality Brachytherapy for Localized Intermediate-to-High-Risk PCa in Japan

In this section, we introduce two clinical trials conducted in Japan to evaluate the efficacy of brachytherapy in patients with intermediate- or high-risk PCa, with and without prolonged hormonal therapy. The high-risk group was supplemented with external beam irradiation as part of the tri-modality therapy. The studies also assessed how a combination of these therapies should be implemented for each risk group, a topic that remains unclear. D'Amico et al. reported the clinical benefits of external beam radiation therapy combined with six months of hormone therapy for localized intermediate- and high-risk PCa [11]. However, no studies have examined the effectiveness of long-term hormone therapy combined with brachytherapy to determine the appropriate use of hormone therapy. Regarding hormone therapy, the PROST-QA study assessed the effect of adjuvant hormone therapy on the QOL parameters for patients with PCa [12]. It was shown that significantly worse sexual function, vitality, fatigue, weight gain, gynecomastia, depression, and hot flashes occurred after two years of hormone therapy, whereas less than one year of hormone therapy demonstrated a significant decrease in these symptoms [12]. Based on these findings, a clinical trial was planned in Japan to find the answer to the clinical question of whether or not combination therapy with nine-month hormone therapy is effective. One study, "Seed and hormone for intermediate-risk prostate cancer (SHIP) 0804", was designed to examine this issue. SHIP 0804 is a phase III, multicenter, randomized controlled study conducted in Japan that compared brachytherapy with short- and long-term hormonal treatment in patients with intermediate-risk PCa. Both groups received neoadjuvant hormonal therapy for three months. The patients in one group received no further therapy, whereas their counterparts underwent a nine-month adjuvant hormonal therapy. The results of the SHIP 0804 trial could identify the rationale for hormonal therapy in patients with intermediate-risk PCa undergoing brachytherapy [13]. The planned 10-year follow-up in SHIP0804 after brachytherapy has just been completed, and the data are being analyzed accordingly. Furthermore, in the treatment of high-risk PCa, single treatment options such

as radical prostatectomy, external beam radiation therapy, and brachytherapy as the initial treatment could lead to treatment failure, including local and biochemical recurrence [14,15]. There is a need for more effective combined treatments with fewer side effects. Since 2010, tri-modality treatment methods such as brachytherapy, external beam radiation therapy, and hormonal therapy have been reported [16–18]. The effectiveness of external radiation therapy combined with hormone therapy for high-risk PCa has also been previously reported [19,20]. Nevertheless, the optimal hormone therapy for tri-modality therapy has not been verified. Another trial titled "Tri-Modality therapy with I-125 brachytherapy, external beam radiation therapy, and short-or long-term hormone therapy for high-risk localized prostate cancer (TRIP)" has also matured. This phase III, multicenter, randomized controlled trial also evaluated the efficacy of brachytherapy combined with external beam irradiation with shorter- versus long-term hormonal therapy in patients with high-risk PCa. The manuscript has been submitted to another journal and is currently under peer review [21].

## 5. Summary of Implantation Methods and Seed Type in Brachytherapy

Brachytherapy requires at least 25 cases to achieve a standard level of implantation proficiency, including needling and dosimetry skills [22]. There are many tips detailing the position of lithotripsy, ingenuity of puncturing through pubis obstacles, adjustment of needle direction using a diddler, fewer punctures to reach the point that the radiation oncologists require, and accurate placement of seeds. In our institutes, approximately 30 cases are required to acquire appropriate experience regarding the standard variations in brachytherapy processes. Notably, communication with radiation oncologists highly skilled in brachytherapy during the procedures can be the key to accomplishing satisfactory outcomes. As mentioned previously, the number of urologists performing brachytherapy tends to be low, even in major institutes. Future studies should focus on ways to address this limitation regarding the number of trained urologists. This will lead to the advantage of reducing the average workload for brachytherapy and passing down specific skills related to the procedure.

There are questions regarding the seed type of brachytherapy, including which seeds lead to better brachytherapy: single or linked strand-type seeds? Linked strand-type seeds have been reported to require 42 min per case of brachytherapy [23]. They also enable more stable and accurate implantation. Furthermore, linked strand-type seeds can be placed outside the prostate capsule, resulting in sufficient radiation dose distribution outside the prostate gland, such as the cT3a or seminal gland, in patients with PCa grade T3b [24,25]. In contrast, single-type seed implants may lead to seed migration, especially when urologists try to implant a single seed on the far distal (apex of the prostate) side. The bloodstream from the needle hole and negative pressure may cause this phenomenon; urologists and radiation oncologists may experience stress when it occurs, resulting in inaccurate implantation and dosimetry distributions. Considering this information, linked strand-type seeds may be the gold standard for implantation.

## 6. Tri-Modality Therapy for Locally Advanced PCa and Adverse Events after Tri-Modality Therapy

The efficacy of tri-modality therapy for patients with intermediate- and high-risk PCa has been sufficiently demonstrated. The ASCENDE-RT study reported that tri-modality therapy was superior to high-dose external beam radiation therapy combined with hormonal therapy for intermediate-to-high-risk PCa. In the study, the 5-, 7-, and 9-year biochemical failure rates were, respectively, 89%, 86%, and 83% in the tri-modality group versus 84%, 75%, and 62% in the hormonal therapy group. The median follow-up was 6.5 years. No significant difference in overall survival was observed between the two treatment modalities [26]. Mari et al. reported the outcomes and unfavorable prognostic factors in patients with PCa stage T3a treated with tri-modal therapy. During a median follow-up of 71 months, the biochemical failure-free survival (BFFS), cancer-specific survival (CSS),

and overall survival rates were 44%, 82%, and 76%, respectively [27]. Another study with a long-term follow-up of 7 years after tri-modality therapy for patients with PCa stage T3 showed an approximately 70% biochemical recurrence-free survival rate [25]. Zhang et al. demonstrated the efficacy of tri-modality therapy in patients with intermediate- and high-risk PCa. The study showed BFFS, CSS, and overall survival rates of 76.6%, 89.1%, and 87.5%, respectively, during the median follow-up of 60 months [28]. Another study reported that patients with high-risk PCa and a Gleason score of 9–10 showed better clinical outcomes, including PCa-specific mortality and longer time to distant metastasis, than the radical prostatectomy group [29]. These results indicate that tri-modality therapy can reach the effective radiation doses required to suppress PSA elevation in patients with intermediate-to-high-risk PCa. The American Brachytherapy Society also recommends androgen deprivation therapy for patients with intermediate-to-high-risk PCa [6]. Table 1 summarizes findings from the studies on tri-modality therapy for locally advanced PCa. Subsequent treatment after the biochemical failure of external beam radiation therapy combined with hormonal therapy may be associated with equivalent overall survival rates in the two modality groups. Regarding hormonal therapy, previous reports showed that high-dose external beam radiation therapy combined with hormonal therapy decreased PSA biochemical recurrence and improved the overall survival of patients with locally advanced or recurrent PCa [30,31]. Therefore, hormonal therapy might be essential to achieve good clinical outcomes in patients with advanced PCa. However, no randomized studies have been conducted to show the effectiveness of hormone therapy on brachytherapy.

**Table 1.** Summary of biochemical recurrence-free and overall survival rates after tri-modality therapy for localized advanced or high-risk PCa. BT: brachytherapy, EBRT: external beam radiation therapy, b-PFS: biochemical progression-free survival, OS: overall survival, ADT: androgen deprivation therapy, MAB: maximal androgen blockade.

| Authors | Patients | Control | Hormonal Therapy | BT | EBRT | b-PFS | OS |
|---|---|---|---|---|---|---|---|
| Mai et al., 2015 [27] | cT3a | n.a. | 24 to 36 months ADT | LDR | 45 Gy irradiation | 44% (5-years) | 76.0% (5-years) |
| Agawal et al., 2018 [25] | cT3a | n.a. | | LDR | | 65.2 (7-years) | 77.9% (7-years) |
| Zhang et al., 2020 [28] | intermediate to high risk Pca | brachytherapy with hormonal therapy | more than 6 months (MAB) | LDR | 45 Gy irradiation | 60.9% vs. 76.6% (median of 43.3 months) | 81.3% vs. 87.5% (median of 43.3 months) |
| Morris et al., 2017 [26] | high risk PCa | 78 Gy of high dose irradiation therapy | 12 months ADT | LDR | pelvic 47 Gy irradiation | 62.0% vs. 83.0% (9-years) | 74.0% vs. 78.0% (9-years) |
| Denham et al., 2018 [31] | locally advancet Pca | short term ADT with radiation therapy alone | another 12 months of ADT | HDR | 66, 70, or 74 Gy | 54.1% vs. 66.0% (10-years) | 67.7% vs. 72.0% (10-years) |

Physicians, including urologists, should realize that patients with PCa and PSA biochemical recurrence exhibit different degrees of disease progression. A previous study reported on a group of patients who showed early disease progression with PSA elevation and suppression of paralleled disease control [32]. That group of patients may have benefitted from PSA suppression therapy. Another patient group showed late disease progression with gradual elevation in PSA levels. PSA suppression therapy had a lower effect on overall survival in that group. Another report by Martin et al. showed that brachytherapy alone may be effective in a specific subgroup of patients with intermediate PCa [33]. Brachytherapy alone potentially delivers a lower radiation dose to the urethra and rectum than brachytherapy combined with external beam radiation therapy [34]. In

clinical practice, patients with PCa who had a large prostate volume, showed PSA levels of >10, and were classified as an intermediate-risk group demonstrated good responses to brachytherapy alone. Therefore, the incidence of external beam radiation-induced dysuria and proctitis was lower in studies in which more patients with intermediate PCa could be treated with brachytherapy alone.

Furthermore, some studies have elucidated the shadow points associated with brachytherapy. Dysuria can occur after brachytherapy combined with external beam radiation therapy [35]. These urinary symptoms generally resolve within 1 year after implantation [6]. Proctitis has been reported to occur after brachytherapy in 2.9–5.7% of patients with PCa [36,37]. In this context, proctitis after brachytherapy for pelvic cancer, including PCa, was associated with older age and a higher radiation dose [38]. Therefore, the injection of a temporary hydrogel in the plane between the prostate and rectum, known as a hydrogel spacer, has been widely used for brachytherapy alone and brachytherapy combined with external beam radiation therapy. The hydrogel spacer is a bioabsorbable gel used to protect the rectum and surrounding tissues during radiation therapy of the prostate. Moroiados et al. reported that patients who received external beam radiation therapy using q hydrogel spacer showed a lower rectal radiation dose and fewer adverse events [39]. The placement of the hydrogel spacer may increase post-void urine volume in the bladder; however, it does not affect the urinary symptom score [40]. The other side effect of brachytherapy is the potential occurrence of erectile dysfunction (ED) after treatment; ED may be associated with both the total dose of external beam radiation and the patient's age [41]. In a real-world scenario, outpatients who underwent brachytherapy stated that they could achieve an erection but experienced less erectile hardness and could achieve orgasm but ejaculated less semen after treatment than before treatment [42].

## 7. Salvage Brachytherapy for PCa

Another useful strategy of LDR brachytherapy for PCa is salvage LDR brachytherapy after external beam radiation. Juanita et al. reported the efficacy of salvage LDR brachytherapy in a phase II clinical trial [43]. In the study, patients received external beam radiation therapy with 74 Gy for 30 months before registration. Patients with favorable- or intermediate-risk PCa with PSA < 20 ng/mL, Gleason score < 7, and clinical T stage T2c or lower were included. The last follow-up duration was 5 years, and the disease-free, biochemical recurrence-free, and overall survival rates were 19%, 46%, and 70%, respectively. Yamada et al. reported the efficacy of salvage brachytherapy [44]. In their study, target lesions were detected using 3 Tesla magnetic resonance imaging (MRI) and prostate biopsy with template methods fused to MRI images. Ultrafocal, hemi-salvage, or whole-salvage brachytherapy was performed according to suspected positive PCa lesions. Biochemical PSA failure was observed in 0/3, 1/5, and 3/5 cases at 48 months of follow-up. The biochemical progression-free survival rate was 75% over 4 years. Therefore, salvage focal brachytherapy may be less invasive for small focal procedures.

## 8. Focal Brachytherapy for PCa

Recently, focal low dose rate (LDR) brachytherapy has been reported for localized PCa [45,46]. Minh-Hanh et al. demonstrated satisfactory five-year biochemical relapse-free, disease-free, and overall survival rates of 96.8%, 79.5%, and 100%, respectively. However, one limitation of that study is that only patients with low-to intermediate-risk PCa were included [45]. Langrey et al. and Laing et al. reported that hemi-prostate gland brachytherapy showed good clinical outcomes, similar to those of whole-gland prostate brachytherapy, in terms of PSA control and overall survival [47,48]. Elliot et al. demonstrated the efficacy of focal LDR brachytherapy. In their study, 26 patients with low-to intermediate-risk PCa were treated with focal brachytherapy, and the data on adverse events and oncological outcomes were retrospectively analyzed. One case of urinary retention and infection was observed in all cases. Nine (37.5%) grade 1 and seven (29.2%) grade 2 urinary dysfunction cases were observed. Eight patients with ED of grade 2 or lower were also observed. A total of 21 patients were evaluated for PCa recurrence using rebiopsy and MRI; none

of the patients had PCa relapse. Only one patient showed PSA recurrence, for which radical prostatectomy was performed [49]. The other report showed less genitourinary toxicity in the focal brachytherapy group than in the whole prostate LDR brachytherapy group [46]. Considering these observations, focal brachytherapy is feasible, with good clinical outcomes and low toxicity. Moreover, randomized studies that compare focal therapy of LDR, focal high-dose-rate brachytherapy, and active surveillance for low-to favorable intermediate-risk PCa are currently ongoing, and their protocols have been published [50]. In that study, 150 patients were randomly enrolled and divided according to treatment options. The primary outcomes were the quality of life after treatment and biochemical recurrence-free survival. Based on the results of that clinical trial, evidence for focal LDR brachytherapy alone for low- to intermediate-risk PCa could be established. It is beneficial for urologists to treat patients with PCa using less invasive procedures.

With regard to new-generation PCa imaging methods, prostate-specific membrane antigen (PSMA) scans using positron emitters such as gallium-68 (Ga), copper-64 (Cu), and fluorine-18 (F), measured with positron emission tomography (PET), could detect PCa lesions more precisely [51]. It may be possible to obtain more accurate images of primary PCa lesions by using a combination of MRI and PSMA-PET, which can help urologists to select treatment options more effectively [52,53]. PSMA-PET shows a small mass of PCa cells in the bone and soft tissue [54]. Body-ablative radiation therapy for oligometastatic lesions can suppress advanced PCa in some patients [54]. These new imaging techniques may make it possible to perform focal therapy, as described. These two methods can be combined to improve the current focal therapy procedures for PCa [55,56]. The utility of deformable registrations of PET/computed tomography (CT) and ultrasound to target PCa was demonstrated in this report. This methodology enables physicians to implant seeds more accurately and to recognize diseased lesions, prostate boundaries, and internal gland shapes, which are sometimes difficult to observe. In the future, a new imaging method comprising PSMA-PET, MRI, and more precise ultrasonography could be applied for focal LDR therapy. In summary, evidence for treatment strategies for intermediate- and high-risk PCa groups and supportive information on focal therapy for low-risk PCa are presented in Table 2.

**Table 2.** Evidence for treatment strategies for intermediate- and high-risk PCa groups and supportive information on focal therapy for low-risk PCa. b-PFS: biochemical progression-free survival, GU: genitourinary, GI: gastrointestinal, N/A: not applicable, G: grade, BT: brachytherapy, EBRT: external beam radiation therapy, and ADT: androgen deprivation therapy.

| Risk of Pca | Randomised Trial, Year, (Ref #) | Suggested Treatment Options | Comparison | b-PFS | GU/GI Toxicities |
|---|---|---|---|---|---|
| low | No randomised trial Supported with several prospective studies, [46,49] | Focal therapy | N/A | 96.8% (5-years) [46] | GU G1: 37.5% GU G2: 29.2% GU G3: 0 % [49] |
| intermediate | RTOG 0232 study, 2023 [10] | BT alone | BT with EBRT vs. BT alone | 88.0% vs. 85.5% (5-years) | G2: 42.8% vs. 25.8% (5-years) G3: 8.2% vs. 3.8% (5-years) |
| high | ASCENDE-RT Trial, 2017 [26,35] | BT boost with pelvic irradiation of 45 Gy under 12 months ADT (Tri-modality therapy) | Tri-modality therapy vs. dose escalated EBRT boost with pelvic irradiation of 45Gy under 12 month ADT | 83.0% vs. 62.0% (9-years) (log-rank $p < 0.001$) | GU G3: 18.4% vs. 5.2% (5-years) GI G3: 8.1% vs. 3.2% (5-years) |

### 9. Current and Future Aspects of Image-Guided Instruments for LDR Needle Puncture

Notably, many image-guided instruments and engineered structures for needle puncture as a high-precision needle insertion strategy have been invented for brachytherapy since the early 2000s. Dai et al. reported that these image-guided instruments were classified as ultrasound-, MRI-, CT-, or fused-image-guided systems [57]. Among these, ultrasound-guided systems have versatility and are the most developed approach due to their familiarity with the procedure among most urologists [58]. However, although MRI-guided systems are superior in terms of image accuracy of the prostate, they should be applied with special devices usable in the magnetic field. MRI-guided systems have the limitation of expensive power sources for piezoelectric actuation [57]. The CT-guided system also has the limitation of radiation exposure for both patients and urologists who deliver seeds manually [59]. Currently, these mechanical instruments assist urologists in performing punctures without a conventional puncture template board, according to the intraoperative plan of radiation oncologists. The priority of these systems is to provide more precise puncture. Other merits of these systems are safety control of puncture and confirmation of needle position. With these systems, urologists and radiation oncologists can perform checks for subsequent procedures [58–60].

Using these robots, LDR-induced complications and unfavorable events could be avoided, including the puncture line of the needle deviating from the site where the radiation source should be placed, patient fatigue caused by longer operation time, increased operator exposure due to extended surgical time, and differences in clinical outcomes between operators. According to the following reports, image-guided instruments for brachytherapy improve clinical outcomes. Podder et al. reported that brachytherapy with image-guided assist instruments achieved more accurate seed placement. Sufficient dosimetric coverage of the prostate was obtained, as in the manual method; however, the number of needles used was reportedly reduced by 30.5% [61]. Furthermore, the same group reported that the number of implanted seeds decreased by 11.8%, and the urethral and rectal radiation doses were reduced, making it possible to perform safer brachytherapy [62]. However, these image-guided instruments for brachytherapy still have limitations. In the case of a narrow public arch, it is difficult to place the seeds in the prostate marginal area; therefore, a technical system is required to advance the needle bending itself. In addition, future technical tasks include developing an automatic needle-loading system and a robotic system that places the seeds.

A research group from China is examining the future directions of brachytherapy regarding prostate needling [63]. This group has invented a mechanical frame to perform brachytherapy needling automatically. These implant systems can help surgeons avoid exposure to radioactive seeds during brachytherapy. Brachytherapy operation time can also be shortened with quick loading and setting steps. Furthermore, human errors during seed counting can also be decreased. Recent progress in artificial intelligence (AI) can also aid implant planning, which is currently being performed by radiation oncologists. In the future, brachytherapy in clinics may be performed automatically by robotic systems guided by AI.

### 10. Conclusions

LDR brachytherapy alone has shown satisfactory clinical outcomes in patients with localized low-to-intermediate PCa. In addition, combination therapy, comprising LDR brachytherapy and external beam radiation and hormone therapies, has achieved beneficial outcomes in patients with localized high-risk PCa. Therefore, physicians should consider utilizing the full potential of LDR brachytherapy in treating PCa. These observed promising results stem from the combined efforts of urologists and radiation oncologists setting precise seeds in the prostate. Currently, LDR brachytherapy is performed by urologists in operating rooms. In the future, novel implantation systems, such as robot-assisted systems, may be

used for LDR implantation. However, the ability of urologists to implant precise seeds to treat patients with PCa should be lauded.

**Author Contributions:** Conceptualization, M.K.; methodology, M.K.; software, S.H.; validation, Y.S., S.K., Y.Y., Y.O. and H.T.; formal analysis, G.K.; investigation, S.T.; resources, M.K.; data curation, M.K.; writing—original draft preparation, M.K.; writing—review and editing, M.K.; visualization, M.K.; supervision, T.K.; project administration, T.I.; funding acquisition, T.K. All authors have read and agreed to the published version of the manuscript.

**Funding:** This research was funded by an Aichi Cancer Center donation for the department of urology.

**Institutional Review Board Statement:** This study did not require an institutional review board statement.

**Informed Consent Statement:** Not applicable.

**Data Availability Statement:** Not applicable.

**Conflicts of Interest:** The authors declare no conflict of interest.

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
