# Peer review of "Clinical Efficacy and Openness to New Challenges of Low Dose Rate Brachytherapy for Prostate Cancer"

_curroncol, doi:10.3390/curroncol30110713_

Round 1

Reviewer 1 Report (Previous Reviewer 1)

Comments and Suggestions for Authors

The authors present a review focussing on LDR brachytherapy for prostate cancer. The manuscript has been improved in the current version. However, the structure still needs to be considerably improved.

Major points:

The manuscript must focus on published studies (part 3 does not report relevant information).

Treatment technique and potential improvements should be addressed in a single chapter.

Search strategy for included manuscripts must be reported.

Evidence for treatment strategies for specific risk groups and focal therapy should be presented in tables, including biochemical recurrence and toxicity, comparison with other treatment options (especially randomized studies).

Example: 

Combined modality treatment is suggested for intermediate risk cancer - what is the evidence? RTOG 0232 study supports LDR brachytherapy alone.

What is the evidence for hormonal therapy? Studies using external beam radiotherapy with hormonal therapy are nor adequate to support evidence for LDR brachytherapy.

Comments on the Quality of English Language

-

Author Response

As reviewer 1 indicated we explained relevant reported study that were origin of the idea for clinical trials.

Line 135 to 175:

In this section, we introduce two clinical trials conducted in Japan to evaluate the efficacy of brachytherapy in patients with intermediate- or high-risk PCa, with and without prolonged hormonal therapy. The high-risk group was supplemented with external beam irradiation as part of the tri-modality therapy. The studies also assessed how a combination of these therapies should be implemented for each risk group, a topic that remains unclear. D’Amico et al. reported the clinical benefits of external beam radiation therapy combined with six months of hormone therapy for localized intermediate- and high-risk PCa. (11) However, no studies have examined the effectiveness of long-term hormone therapy combined with brachytherapy to determine the appropriate use of hormone therapy. Regarding hormone therapy, the PROST-QA study assessed the effect on QOL parameters by adjuvant hormone therapy for patients with PCa. (12) It has been shown that significantly worse sexual function, vitality, fatigue, weight gain, gynecomastia, depression, and hot flashes occur after two years of hormone therapy, whereas less than one year of hormone therapy has demonstrated a significant decrease in these symptoms. (12) Based on these findings, a clinical trial was planned in Japan to find the answer to the clinical question of whether or not combination therapy with nine-month hormone therapy is effective. One study, “Seed and hormone for intermediate-risk prostate cancer (SHIP) 0804,” was designed to examine this issue. SHIP 0804 is a phase III, multicenter, randomized controlled study conducted in Japan that will compare brachytherapy with short- and long-term hormonal treatment in patients with intermediate-risk PCa. Both groups received neoadjuvant hormonal therapy for three months. Patients in one group received no further therapy, whereas their counterparts underwent nine -month adjuvant hormonal therapy. The results of the SHIP 0804 trial could identify the rationale for hormonal therapy in patients with intermediate-risk PCa undergoing brachytherapy. (13) The planned 10-year follow-up in SHIP0804 after brachytherapy has just been completed, and data are being analyzed accordingly. Furthermore, in the treatment of high-risk PCa, single treatment options such as radical prostatectomy, external beam radiation therapy, and brachytherapy as the initial treatment could lead to treatment failure, including local and biochemical recurrence. (14, 15) There is a need for more effective combined treatments with fewer side effects. Since 2010, tri-modality treatment methods such as brachytherapy, external beam radiation therapy, and hormonal therapy have been reported. (16-18) The effectiveness of external radiation therapy combined with hormone therapy for high-risk PCa has been previously reported. (19, 20) Nevertheless, the optimal hormone therapy for tri-modality therapy has not been verified. Another trial titled “Tri-Modality therapy with I-125 brachytherapy, external beam radiation therapy, and short-or long-term hormone therapy for high-risk localized prostate cancer (TRIP)” has also matured. This phase III, multicenter, randomized controlled trial also evaluated the efficacy of brachytherapy combined with external beam irradiation with shorter- versus long-term hormonal therapy in patients with high-risk PCa. This manuscript has been submitted to another journal and is currently under peer review. (21)

As reviewer 1 indicated that treatment technique and potential improvements should be addressed in a single chapter, therefore we divided previous chapter 5 “5. Advances in tri-modality therapy, salvage brachytherapy, focal therapy, and future robotic brachytherapy development: openness to new challenging procedures of LDR brachytherapy” into “Tri-modality therapy for locally advanced PCa and adverse events after tri-modality therapy”, “Salvage brachytherapy for PCa”, “Focal brachytherapy for PCa” and “Current image-guided instruments for LDR’s needle puncture and future aspects” as Chapter 6, 7, 8, and 9, respectively. With these modification, chapter number of “Conclusions” changed into “10.”.

As reviewer 1 indicated we added search strategy for included in manuscript as “2. Materials and Methods” section shown in below.

Line 82 to 99:

  1. Materials and Methods

To perform this study, a review of current LDR brachytherapy for prostate cancer, our group conducted a comprehensive search using different databases. Different databases included Pubmed, Springer, Elsevier Science Direct, and even Web of Science. The publication years of these original articles from different databases range from 2000 to 2023, taking into consideration the validity. The selected papers were only reported in English and the searches were conducted using the keywords shown below. The keywords were “prostate cancer”, “low-dose-rate brachytherapy”, “brachytherapy for prostate cancer”, “tri-modality therapy for prostate cancer”, “LDR brachytherapy for intermediate risk prostate cancer”, “LDR brachytherapy for high risk prostate cancer,” “salvage brachytherapy for prostate cancer,” “focal brachytherapy for prostate cancer,” and “image-guided instrument for prostate brachytherapy.” Duplicated papers were excluded. During the process of checking the contents of the papers, they were screened to exclude papers unrelated to this review and reports that were not full text were excluded, either. There are several criteria for inclusion. The emphasis is on original articles of clinical research about LDR brachytherapy for prostate cancer or review articles. The exclusion criteria were mainly focusing on different aspect of LDR brachytherapy for prostate cancer, articles containing information unrelated to LDR for prostate cancer, and reports that could not be reached as full text articles.

As reviewer 1 indicated we described additional table 2 that showed summary of treatment strategies for specific risk groups and focal therapy especially regarding to biochemical recurrence and toxicity among the treatment options.

In this table 2, we have quoted randomized studies on the treatment strategies for intermediate- and high- risk PCa and supportive information for low-risk PCa

Line 350 to 352:

In summary, evidence for treatment strategies for intermediate- and high-risk PCa groups and supportive information on focal therapy for low-risk PCa are presented in Table 2.

Line 354 to 357:

Table 2. Evidence for treatment strategies for intermediate- and high-risk PCa groups and supportive information on focal therapy for low-risk PCa. b-PFS: biochemical progression-free survival, GU: genitourinary, GI: gastrointestinal, N/A: not applicable, G: grade, BT: brachytherapy, EBRT: external beam radiation therapy, ADT: Androgen deprivation therapy

Along with this table 2, we explained evidence of RTOG 0232 study as treatment strategy for intermediate PCa in section 3. “Clinical outcomes of localized low-to-intermediate Pca treatment” with sentences shown in below.

Line 123 to 129

One randomized study (RTOG0232) for intermediate-risk PCa, particularly in the GS6 and PSA 10-20 and GS7 and PSA <10 groups, there was no significant difference in biochemical recurrence free survival (biochemical failure) between brachytherapy combined with external beam radiation therapy and brachytherapy alone. (88.0% vs. 85.5% at 5 years) However, there was a significant difference in grade 2 or higher genitourinary (GU) or gastrointestinal (GI). (42.8% vs 25.8% in G2, 8.2% vs 3.8% in G3 at 5 years) (10) 

As for focal therapy of LDR, randomized studies that comparing focal therapy of LDR, focal high-dose-rate brachytherapy and active surveillance for low to favourable: intermediate-risk PCa are currently on going and its protocol were published. [Ref # 51: Janusas J. et al. 2023]

We explained this study in section 8. “Focal brachytherapy for PCa” shown in below.

Line 327 to 334:

Moreover, randomized studies that compare focal therapy of LDR, focal high-dose-rate brachytherapy, and active surveillance for low-to favorable intermediate-risk PCa are currently ongoing, and their protocols have been published. (50) In this study, 150 patients were randomly enrolled and divided according to treatment options. The primary outcomes were the quality of life after treatment and biochemical recurrence-free survival. Based on the results of this clinical trial, evidence for focal LDR brachytherapy alone for low-to intermediate-risk PCa could be established. It is beneficial for urologists to treat patients with PCa using less invasive procedures.

As reviewer 1 indicated it is logically correct that studies using external beam radiotherapy with hormonal therapy are nor adequate to support evidence for LDR brachytherapy.

Therefore, we added the explanation of this limitation that describe no randomized studies were conducted to show the effectiveness of hormone therapy on brachytherapy.

Line 232 to 235:

Therefore, hormonal therapy might be essential to achieve good clinical outcomes in patients with advanced PCa. However, no randomized studies were conducted to show the effectiveness of hormone therapy on brachytherapy.

Reviewer 2 Report (Previous Reviewer 2)

Comments and Suggestions for Authors I  would like to acknowledge the work of the authors reviewing their paper and I now  believe that this version of the  manuscript has been significantly improved and now warrants publication.

Author Response

Thank you so much for your comment and suggestions on this manuscript.

I will correct the manuscript for other reviewer and make it more sophisticated for publication.

Thank you for your understanding of this current version of the manuscript.

Manabu Kato,

Aichi Cancer Center, Urology

Round 2

Reviewer 1 Report (Previous Reviewer 1)

Comments and Suggestions for Authors

This is a general descriptive manuscript concerning various LDR brachytherapy topics, giving an overview to readers without deeper insite in this topic.

This manuscript is a resubmission of an earlier submission. The following is a list of the peer review reports and author responses from that submission.

Round 1

Reviewer 1 Report

Comments and Suggestions for Authors

The manuscript includes an outcome evaluation of an institute and several other aspects of LDR brachytherapy. It lacks a good focus.

A manuscript should either be a study focused on a patient group or a review focussing on a specific group or a specific treatment. 

Comments on the Quality of English Language

There are some minor language problems, as e.g. in the abstract "...with regards to the late 90% of 10 years..."

Author Response

Thank you so much for reviewing our manuscript. 

I corrected point by point as reviewer1 indicated.

Reviewer 1’s comments

The manuscript includes an outcome evaluation of an institute and several other aspects of LDR brachytherapy. It lacks a good focus.

A manuscript should either be a study focused on a patient group or a review focussing on a specific group or a specific treatment. 

There are some minor language problems, as e.g. in the abstract "...with regards to the late 90% of 10 years..."

As reviewer 1 indicated, we have excluded clinical outcomes and corrected the manuscript to focus on a review of a specific each PCa group treated with LDR and LDR combination therapy. Specific LDR related treatment were also mentioned in the manuscript.

In particular, we omitted "Our institutes demonstrated acceptable clinical outcomes with regards to the late 90% of 10 years prostate-specific antigen (PSA) biochemical recurrence-free rate" from abstract part, previously on line 25-26. We also omitted "In the first part of this review, medical treatment flow and clinical outcomes of patients who underwent LDR brachytherapy in two of our institutes are discussed." from introduction part, previously on line 62-63.

According to these changes, "In the second part, clinical outcomes of LDR brachytherapy in multiple institutes in Japan have been shown." was changed into "In the first part, clinical outcomes of LDR brachytherapy in multiple institutes in Japan have been shown." In the Introduction. Along this, "In the third part, clinical outcomes of patients with localized low-to-intermediate PCa have been described. In the fourth part, clinical trials investigating multi-modal brachytherapy for localized intermediate-to-high risk PCa in Japan have been discussed." was changed into "In the second part, clinical outcomes of patients with localized low-to-intermediate PCa have been described. In the third part, clinical trials investigating multi-modal brachytherapy for localized intermediate-to-high risk PCa in Japan have been discussed." in the introduction, either.

In addition, we excluded "2. Medical treatment flow and clinical outcomes of LDR brachytherapy in our institutes" section, "2.1. Characteristic of patients with PCa in our institutes", "2.2. Parameters of radiation dose post-plan", "2.3. Clinical outcome after brachytherapy for PCa", "Table 1. Characteristics of patient treated with brachytherapy in our institutes.", "Table 2. Dosimetric parameters of patients treated with brachytherapy in our institutes.", "Figure 1. Overall survival rates following LDR alone and LDR with extra-beam radiotherapy.", "Figure 2. Biological recurrence-free rates following LDR alone and LDR with extra-beam radiotherapy.", "Figure 3. Biological recurrence-free rates of all patients sorted according to PCa risk classifications.", and "Figure 4. Rate of prostitis occurrence after LDR.". As followed, reference of (6) that is McNeely LK, Stone NN, Presser J, Chircus JH, Stock RG. Influence of prostate volume on dosimetry results in real-time 125I seed implantation. Int J Radiat Oncol Biol Phys. 2004;58:292-9. was omitted.

We also changed the sentences in "2. Clinical outcomes for localized low to intermediate Pca" on line 92-94, into "Clinical outcomes for localized low to intermediate Pca in our institutes also reflected the general outcomes of brachytherapy in Japanese patients with low-to-intermediate PCa, as reported in a previous review. (7) (5)".

The following paragraph's numbers were also changed as shown below.

2. Clinical outcomes for localized low to intermediate Pca

3. Clinical trial of multi-modality brachytherapy for localized intermediate to high risk PCa in Japan

4. Summary of implantation methods and seed type in brachytherapy

5. Advances in tri-modality therapy, salvage brachytherapy, focal therapy, and future robotic brachytherapy development: openness to new challenging procedures of LDR brachytherapy

5.1. Tri-modality therapy for locally advanced PCa and adverse events after tri-modality therapy

Table 1 summarizes findings from the studies on tri-modality therapy for locally advanced PCa.

Table 1. Summary of biochemical recurrence-free rates and overall survival rates following tri-modality therapy for localized advanced or high-risk PCa.

5.2. Salvage brachytherapy for PCa]

5.3. Focal brachytherapy therapy for PCa

5.4. Future automatic brachytherapy equipment development

6. Conclusions

The reference numbers were corrected as we omitted previous number (6) reference. As a result, the reference numbers have changed into revised form.

As reviewer 1 indicated on this manuscript some English problem, we asked English correction company again to get the manuscript more sophisticated English version.

In this process, I have corrected "real world" into "real-world" on line 220, "I this large review" into "One large review" on line 85-86.

Reviewer 2 Report

Comments and Suggestions for Authors

I am afraid that I am unable to assess this work for the following reasons::

There is a huge problem with this paper regarding the use of the english language

(The authors state that “English correction was conducted by editage. Pag 12, line 377)

There are also major issues with the use of technical and medical terms refering to the brachytherapy procedures and a huge confusing regarding the medical specialists treating patients with radiation.

These issues clearly affect the clarity and readability of the document and here are just a few (of many) examples to justify my decision:

Page 1, line 31:

Combined use of LDR brachytherapy with other therapeutical modalities, such as extra-beam radiation and androgen deprivation therapy…...

I believe that this is a mistake, surely the authors mean external radiation (or external beam radiation)

Throughout the paper, there is confusion regarding the medical specialists in charge of the procedure:

Pag 3, line 102:

The brachytherapy procedure was performed by clinicians with precise training and education, as per the guidelines reported by Stone et al. (6)

I have read the paper by Stone (ref. 6) and what it says is: ….community hospital practitioners trained in real-time US guidance”…….

Well, clearly a doctor who specializes in treating patients with radiation therapy is a radiation oncologist: I have checked and according to the website of National cancer Center Hospital in Japan, their Dep. of Radiation Oncology treats the largest number of radiation oncology patients in Japan, with a team including radiation oncologists, radiation technologists, and medical physicists

https://www.ncc.go.jp/en/ncch/clinic/radiation_oncology/index.html

pag 11. line 345:

One group from China is examining the future directions of brachytherapy regarding prostate needling

Do the authors mean “high-precision needle insertion strategy” as the authors they quote ( ref 43)?

Comments on the Quality of English Language

English is very difficult to understand and the technical terms are sometimes incomprehensible

Author Response

Thank you so much for reviewing our manuscript.

I have asked special department of editage to correct English language. I have also corrected point by point for reviewer2's comments.

As Reviewer 2 indicated, we ask English editing company again to correct English language.

As Reviewer 2 indicated, we corrected all expressions of "extra-beam radiation" into "external beam radiation" on line 27, 42, 55, 67, 91, 96, 110, 123, 126, 161, 182, 185, 198, 201, 205, 212, 214, 219, 231, and 321.

Reviewer 2 indicated that clearly a doctor who specializes in treating patients with radiation therapy is a radiation oncologist that is also mentioned on the web site of National cancer Center Hospital in Japan.

Thus, to avoid confusions we have corrected clinicians into radiation oncologists who are in charge of LDR brachytherapy for real on line 26, 70, 72, 97, 102, 134, 136, 152, 289, 292, 315, and 324..

The sentence of Pag 3, line 102, "The brachytherapy procedure was performed by clinicians with precise training and education, as per the guidelines reported by Stone et al. (6)" was omitted for the indication of reviewer 1. As a result, the reference numbers have changed into revised form.

English is very difficult to understood and the technical terms are sometimes incomprehensible

As reviewer 2 indicated, our English expressions were confusing and might lead misunderstandings. Thus, we corrected reviewer 2's indicated points and let the manuscript to ask for English correction to become more sophisticated and well understanding.

One group from China is examining the future directions of brachytherapy regarding prostate needling. Do the authors mean “high-precision needle insertion strategy” as the authors they quote ( ref 43)?

As reviewer2 indicated, we added the sentence to explain more clearly in this session, “Notably, many image-guided instruments and engineered structures for needle puncture as high-precision needle insertion strategy have been invented for brachytherapy since the early 2000s.” on line 278 to 280.

Reviewer 3 Report

Comments and Suggestions for Authors

Thank you for the submitted manuscript.

I found it difficult to understand the purpose of your manuscript. I would suggest separating it into two manuscripts. The retrospective study should be properly presented as a clinical study; introduction, methodology, results, and discussion.

The second part as a review of the technique and future direction could be more organized and extended.

The manuscript in its present form is vague and not complete in both aspects.

A professional English-language review may help in rephrasing some parts of the manuscript as they are vague and hard to understand in their current form.

Terminology is very important; there is nothing called biological recurrence, I think the authors mean biochemical. The term extra-beam is also wrong, it should be external-beam radiation treatment (EBRT). Also, irradiation is not equal to EBRT and the term should be corrected throughout the manuscript.

Abbreviations should be stated. Examples, are UD30, RV100, and Table 2.

Figure 4 is wrong, is it about prostatitis or proctitis? please correct the citation in line 144.

line 167, institutes, correct!

Many parts are lacking citations! examples: 1264-268, 270-272, 280-281., 290-293.

172-773: vague.

Section 3. needs an extensive rephrasing.

line 282: SpaceOAR is a trade name. it should be corrected.

169, and 326-27: correct spelling and grammar

Comments on the Quality of English Language

A professional English-language review is needed.

Author Response

Thank you so much for revewing our manuscript.

Reviewer 3's commnets

I found it difficult to understand the purpose of your manuscript. I would suggest separating it into two manuscripts. The retrospective study should be properly presented as a clinical study; introduction, methodology, results, and discussion.

The second part as a review of the technique and future direction could be more organized and extended. The manuscript in its present form is vague and not complete in both aspects.

As reviewer 3 indicated, I omitted first part of retrospective study to make this manuscript as a review work. Then I extended the part of  "5.4 Current image-guided instruments for LDR’s needle puncture and future aspects" as shown in below.

5.4 Current image-guided instruments for LDR’s needle puncture and future aspects

Notably, many image-guided instruments and engineered structures for needle puncture as high-precision needle insertion strategy have been invented for brachytherapy since the early 2000s. Dai et al. reported that these image-guided instruments were classified as ultrasound-, MRI-, CT-, or fused-image-guided systems. (46) Among these, ultrasound-guided systems have versatility and are the most developing approach due to the accustomed image procedure for most urologists. (47) However, although MRI-guided systems were superior in image accuracy of the prostate, they should be applied with special devices usable in the magnetic field. MRI-guided systems have a limitation of expensive power sources of piezoelectric actuation. (46) CT-guided system also has a limitation of radiation exposure for both patients and urologists who deliver seeds manually. (48) Currently, these mechanical instruments assist urologists in performing punctures without a conventional puncture template board, according to the intraoperative plan of radiation oncologists. The priority of these systems is to provide more precise puncture. Other merits of these systems were safety control of puncture with confirmation of needle position. With these systems, urologists and radiation oncologists can take checks for subsequent procedures. (48) (49) (47)

Using these robots, LDR-induced complications and unfavorable events could be avoided, including the puncture line of the needle deviating from the site where the radiation source should be placed, patient fatigue caused by longer operation time, increased operator exposure due to extended surgical time, and differences in clinical outcomes between operators. According to the following reports, image-guided instruments for brachytherapy improved clinical outcomes. Podder et al. reported that brachytherapy with image-guided assist instruments achieved more accurate seed placement. Sufficient dosimetric coverage of the prostate was obtained as in the manual method; however, the number of needles used was reportedly reduced by 30.5%. (50) Furthermore, the same group reported that the number of implanted seeds decreased by 11.8%, and the urethral and rectal radiation doses were reduced, making it possible to perform safer brachytherapy. (51) However, these image-guided instruments for brachytherapy still have limitations. In the case of a narrow public arch, it is difficult to place the seeds in the prostate marginal area; therefore, a technical system is required to advance the needle bending itself. In addition, future technical tasks include developing an automatic needle-loading system and a robotic system that places the seeds.

One group from China is examining the future directions of brachytherapy regarding prostate needling. (52) This group invented a mechanical frame to perform brachytherapy needling automatically. These implant systems can help surgeons avoid exposure to radioactive seeds during brachytherapy. The brachytherapy operation time can also be shortened with quick loading and setting steps. Furthermore, human errors during seed counting can also be decreased. Recent progress in artificial intelligence (AI) can also aid implant planning, which is currently being performed by radiation oncologists. In the future, brachytherapy in clinics may be performed automatically by robotic systems guided by AI.

A professional English-language review may help in rephrasing some parts of the manuscript as they are vague and hard to understand in their current form.

As reviewer 3 indicated, we got this manuscript checked by English-language review again to rephrase many parts of the manuscript.

Terminology is very important; there is nothing called biological recurrence, I think the authors mean biochemical. The term extra-beam is also wrong, it should be external-beam radiation treatment (EBRT). Also, irradiation is not equal to EBRT and the term should be corrected throughout the manuscript.

As reviewer 3 indicated, we have corrected biological recurrence into "biochemical recurrence" on line 86, 95, 162, 168, 171, 182, 186, 189, 223, 224, 234, 239, 240, and 245.

The wrong term of "extra-beam" were corrected into "external beam", on line on line 27, 42, 55, 67, 91, 96, 110, 123, 126, 161, 182, 185, 198, 201, 205, 212, 214, 219, 231, and 321.

The term "irradiation" were also corrected into "external beam radiation".

Abbreviations should be stated. Examples, are UD30, RV100, and Table 2.

As reviewer 3 indicated, abbreviations have been stated shown below. In fact, the table using UD30, RV100 was omitted.

Table 1. Summary of biochemical recurrence-free rates and overall survival rates following tri-modality therapy for localized advanced or high-risk PCa. b-PFS: biochemical progression free survival, OS: Overall survival, ADT: Androgen deprivation therapy, MAB: Maximal androgen blockade

Figure 4 is wrong, is it about prostatitis or proctitis? please correct the citation in line 144.

As reviewer3 pointed, Figure4 was wrong. It was about proctitis.

Figure 4 was omitted to make this manuscript as a review work.

line 167, institutes, correct!

As reviewer3 indicated, "in all clinical institute" was changed into "in all clinical institutes" on line 84.

Many parts are lacking citations! examples: 1264-268, 270-272, 280-281., 290-293.

As reviewer 3 indicated, citations were added to line 190-192, 196-198, 208-209, and 219-222.

172-773: vague.

As reviewer 3 indicated, we corrected the sentence as shown below to make it clear.

On line 94-97

Therefore, the high biochemical recurrence-free survival rate indicated that low-to-intermediate-risk PCa could be controlled using LDR alone or LDR combined with external radiation therapy under appropriate treatment selection.

Section 3. needs an extensive rephrasing.

As reviewer 3 indicated, the section 2 (changed from 3) were rephrased shown below.

  1. Clinical outcomes of localized low-to-intermediate Pca treatment

Brachytherapy is considered an effective treatment option for patients with localized PCa. (https://www.auanet.org/guidelines-and-quality/guidelines/clinically-localized-prostate-cancer-aua/astro-guideline-2022) American Brachytherapy Society has stated that brachytherapy is a convenient, effective, and well-acceptable treatment for localized PCa. (6) However, despite its advantages, brachytherapy cannot be performed in all clinical institutes, leading to the impression of this treatment as a “minor” option. One large review from Japan showed a median follow-up duration of 75 months and 7-year biochemical recurrence-free survival rates of 98%, 93%, and 81% in patients with low-, intermediate-, and high-risk PCa, respectively. (5) (7) The indications for combination therapy in the study were as follows: low-, intermediate-, and high-risk PCa should be treated with brachytherapy alone, brachytherapy combined with irradiation therapy, and brachytherapy combined with neoadjuvant androgen deprivation and external radiation therapy, respectively. (5) (7) Clinical outcomes of localized low-to-intermediate Pca treatment in our institutes were consistent with the general outcomes of brachytherapy for low-to-intermediate PCa (data not shown), as reported in these reviews. (5) (7) Therefore, the high biochemical recurrence-free survival rate indicated that low-to-intermediate-risk PCa could be controlled using LDR alone or LDR combined with external radiation therapy under appropriate treatment selection. In addition, most Japanese institutes employ radiation oncologists to decide the seed implant position before initiating needle puncture. The seeds were repositioned during the puncture and implantation procedures. This dynamic dose calculation method might produce sufficient dose distribution and good clinical outcomes (8) compared with other procedures in which urologists puncture the prostate before planning seed placement, which is mainly led by radiation oncologists. (9) As mentioned above, despite some unpopularity and limited institutional procedure, strong evidence of the unchangeability and stability of brachytherapy for low-to-intermediate-risk PCa was demonstrated.

line 282: SpaceOAR is a trade name. it should be corrected.

As reviewer 3 indicated, line 209-212: Therefore, the injection of a temporary hydrogel in the plane between the prostate and rectum, known as a hydrogel spacer, has been widely used for brachytherapy alone and brachytherapy combined with external beam radiation therapy. 

Hydrogel spacer were used for explanation on line 212, 215, and 216.

169, and 326-27: correct spelling and grammar

As reviewer 3 indicated, we corrected spelling and grammer

On line 85-88 (previous line 169),

One large review from Japan showed a median follow-up duration of 75 months and 7-year biochemical recurrence-free survival rates of 98%, 93%, and 81% in patients with low-, intermediate-, and high-risk PCa, respectively. (5) (7) 

On line 260-261 (previous line 326-327) 

Considering these observations, focal brachytherapy is feasible, with good clinical outcomes and less toxicity.